# *Sinorhizobium meliloti* GR4 Produces Chromosomal- and pSymA-Encoded Type IVc Pili That Influence the Interaction with Alfalfa Plants

**DOI:** 10.3390/plants13050628

**Published:** 2024-02-25

**Authors:** Cristina Carvia-Hermoso, Virginia Cuéllar, Lydia M. Bernabéu-Roda, Pieter van Dillewijn, María J. Soto

**Affiliations:** Department of Biotechnology and Environmental Protection, Estación Experimental del Zaidín, CSIC, 18008 Granada, Spain; cristina.carvia@eez.csic.es (C.C.-H.); virginia.cuellar@eez.csic.es (V.C.); lydia.bernabeu@eez.csic.es (L.M.B.-R.)

**Keywords:** Tad/Flp pili, *Rhizobium*, plant colonization, nodulation, competitive ability

## Abstract

Type IVc Pili (T4cP), also known as Tad or Flp pili, are long thin microbial filaments that are made up of small-sized pilins. These appendages serve different functions in bacteria, including attachment, biofilm formation, surface sensing, motility, and host colonization. Despite their relevant role in diverse microbial lifestyles, knowledge about T4cP in bacteria that establish symbiosis with legumes, collectively referred to as rhizobia, is still limited. *Sinorhizobium meliloti* contains two clusters of T4cP-related genes: *flp-1* and *flp-2*, which are located on the chromosome and the pSymA megaplasmid, respectively. Bundle-forming pili associated with *flp-1* are involved in the competitive nodulation of alfalfa plants, but the role of *flp-2* remains elusive. In this work, we have performed a comprehensive bioinformatic analysis of T4cP genes in the highly competitive *S. meliloti* GR4 strain and investigated the role of its *flp* clusters in pilus biogenesis, motility, and in the interaction with alfalfa. Single and double *flp*-cluster mutants were constructed on the wild-type genetic background as well as in a flagellaless derivative strain. Our data demonstrate that both chromosomal and pSymA *flp* clusters are functional in pili biogenesis and contribute to surface translocation and nodule formation efficiency in GR4. In this strain, the presence of *flp-1* in the absence of *flp-2* reduces the competitiveness for nodule occupation.

## 1. Introduction

*Sinorhizobium meliloti* is an alpha-proteobacterium that can live as a saprophyte in the soil but can also establish nitrogen-fixing symbiosis with alfalfa plants [1]. Successful establishment of this symbiosis results in the formation of root nodules, which are infected by bacteria that fix atmospheric dinitrogen into ammonia for direct use by the plant. The progress of this interaction entails different phases and a sophisticated molecular dialogue between the legume host and the rhizobial microsymbiont [1,2,3]. An early and crucial step for the establishment of symbiosis is the bacterial colonization of plant roots, a process that is initiated by the movement of rhizobia toward the plant root and is followed by bacterial attachment to the root surface and the development of a microbial community or biofilm [4,5].

Rhizobial motility is not essential for nodulation or nitrogen fixation but is an important trait for root colonization and infection, which eventually could influence nodulation efficiency and competitiveness [6,7,8,9,10,11]. Rhizobial competitiveness (i.e., the ability of a given strain to form nodules in the presence of a second rhizobial strain) is a complex trait that plays a fundamental role in the success of inoculants used as biofertilizers for legumes [12,13,14]. Therefore, the identification and characterization of components and regulatory mechanisms that govern rhizobial motility might contribute to the design of elite inoculants and the development of sustainable agriculture. *S. meliloti* can move using different mechanisms. The movement of bacteria in aqueous environments or swimming is absolutely reliant on flagellar action, whereas translocation of the alfalfa microsymbiont over surfaces can be mediated by both flagella-dependent and –independent mechanisms [10,15,16,17,18]. Swarming is a flagella-driven type of motility characterized by the rapid and coordinated migration of cells across surfaces [19,20,21]. This motility has been described in *S. meliloti* [10,15,16] and in other rhizobia [22,23,24,25]. *S. meliloti* flagellaless strains can exhibit a mode of surface spreading referred to as sliding [26]. In this case, bacterial movement is the result of expansive forces of cell growth together with the production of surfactants that reduce surface tension (e.g., exopolysaccharides and the amphiphilic siderophore, rhizobactin 1021) [10,16]. Another flagella-independent mode of surface translocation described in many bacteria but not yet in rhizobia is twitching, which is mediated by the extension and retraction of type IV pili (T4P) [27,28]. Like flagella, T4P enable bacteria to not only move but also to attach to surfaces contributing to biofilm formation and host colonization [4,11].

T4P are long, flexible, and elastic bacterial extracellular filaments, which are thinner than flagella and are made up of thousands of protein subunits called pilins [27,29,30,31]. These appendages have the peculiarity of being dynamic as rapid polymerization and depolymerization of pilin subunits carried out by a complex machinery permit the pili to extend and contract. This characteristic is essential for the different functions associated with T4P [32].

The classification of T4P distinguishes three types: T4aP, T4bP, and the Tight adherence (Tad) pili, also known as Flp (Fimbrial low-molecular-weight protein) pili or T4cP [33]. One of the features that differentiates the different types of T4P is their major pilin sequences. Pilins of T4aP range between 150–175 amino acids, T4bP pilins are larger (180–200 amino acids), and T4cP are made up of small-sized pilins (50–80 amino acids) [32,34]. In addition, the genes required for T4aP biogenesis are numerous (more than 40) and are often scattered throughout the bacterial genome. In contrast, the synthesis of T4bP and T4cP requires a smaller number of genes (12–14) and they are usually grouped in a region, sometimes with genetic-island characteristics [32].

The genes encoding dedicated proteins for the polymerization and secretion of T4cP have a very specific and conserved genetic organization, as is the case for the Tad system of *Aggregatibacter actinomycetemcomitans* (formerly *Actinobacillus actinomycetemcomitans* [35,36,37,38,39]), the *cpa* (*Caulobacter* pilus assembly) locus of *Caulobacter vibrioides* (formerly *Caulobacter crescentus* [40]), or the *ctp* (cluster of type IV pili) locus of *Agrobacterium fabrum* C58 (formerly *Agrobacterium tumefaciens* C58 [41]) (Figure 1a). Before the major Flp pilin can be incorporated into the growing T4c pilus, it needs to be processed by a specific prepilin peptidase (TadV/CpaA/CtpB) that removes a leader sequence at a conserved glutamate- and tyrosine-containing Flp motif (GXXXXEY [37]). TadV also participates in the maturation of the TadE and TadF pseudopilins, which share the conserved GXXXXEF sequence at their N-terminal but play an as-yet unclear role in pilus biogenesis [38,42,43]. TadA/CpaF/CtpG is the cytosolic ATPase that catalyzes both the extension and retraction of pili with the help of the inner-membrane platform proteins TadB/CpaG/CtpH and TadC/CpaH/CtpI [30]. TadZ/CpaE/CtpF is a docking protein for the Tad secretion system, which may help to localize pili formation to the poles [43,44,45]. In diderms, T4cP pass through the outer membrane via the secretin RcpA/CpaC/CtpD, which requires the lipoprotein TadD/CpaO that functions as a pilotin, enabling the assembly and correct insertion of the secretin into the outer membrane [43,46]. RcpB/CpaD/CtpE and RcpC/CpaB/CtpC probably form a complex with RcpA [39] and possibly are also important for the assembly and stability of the secretin complex [38].

T4cP genes are widely distributed amongst bacteria, indicating that they confer significant adaptive advantages [32,38]. They have been involved in diverse functions such as surface sensing, motility, cell adhesion, biofilm formation, host colonization, virulence, and even bacterial predation [30,32,47]. Concerning the interaction with plant hosts, T4cP have been involved in the virulence of *Ralstonia solanacearum* and *Pectobacterium atrosepticum* [48,49]. In ‘*Candidatus* Liberibacter asiaticus’, the causal agent of citrus greening disease, Tad pili contribute to the colonization of the insect vector and therefore to disease propagation [50].

There is very little information on T4cP in rhizobia in general, and in *S. meliloti* in particular. The first indication of pilus-like structures in rhizobia and their implication in root attachment was described in the soybean symbiont *Bradyrhizobium diazoefficiens* (formerly *Bradyrhizobium japonicum*) by Vesper and Bauer (1986) [51]. Information from genome sequencing has revealed that the majority of rhizobial species carry chromosomal gene clusters putatively encoding T4cP, and in many cases, additional truncated clusters have also been found [11,52,53,54,55,56]. In *B. diazoefficiens* USDA 110, additional *tadGEF* genes were found away from the two main T4cP clusters of *tad* and *cpa* genes, and their loss affected the adhesion to soybean roots and caused delayed root infection [53]. In *S. meliloti* Rm1021, two clusters of genes that encode different components of T4cP have been identified [56]. The *flp-1* cluster is located on the chromosome, while the second cluster, *flp-2*, is located on the symbiotic megaplasmid pSymA and appears to be truncated in Rm1021. Deletion of *pilA1*, which encodes the major pilin of the *flp-1* cluster, abolishes pili formation and reduces the competitive ability of Rm1021 to nodulate alfalfa [56]. However, data available up to now do not implicate these appendages with any type of motility in *S. meliloti* [56], and the role of each *flp* cluster in the nodulation process still remains unclear.

*S. meliloti* GR4 was isolated as a predominant rhizobial strain from an agricultural field in Granada (Spain) [57]. This strain exhibits relevant genomic and phenotypic differences with respect to the well-known Rm1021. GR4 contains two accessory plasmids [58], and compared with other *S. meliloti* strains, including Rm1021, it is highly competitive for the nodulation of alfalfa [13]. The molecular bases responsible for the high competitive ability of GR4 are not fully understood [13]. It has been shown that GR4 exhibits increased biofilm formation on both abiotic and root surfaces compared with Rm1021, which could account for its better symbiotic performance [9]. Yet the role of T4cP in the root-colonization ability and symbiotic behavior of GR4 has not been investigated. In this work, we have performed a comprehensive bioinformatic analysis of T4cP genes in GR4 and investigated the role of its *flp-1* and *flp-2* clusters in pilus biogenesis, motility and in the interaction with alfalfa plants by using single and double *flp*-cluster deletion mutants. Our data demonstrate that both chromosomal and pSymA *flp* clusters are functional in pili biogenesis. These appendages contribute to the surface translocation exhibited by *S. meliloti* GR4 and influence its symbiotic fitness.

## 2. Results

### 2.1. In Silico Analyses of Type IVc Pili (T4cP) Genes of S. meliloti GR4

To identify T4cP-related genes in the *S. meliloti* GR4 strain, its genome was scanned for loci similar to *tad*/*cpa/ctp* genes (Figure 1, Table 1). We found a total of 29 genes, with 16 of them located on the chromosome and 13 on the pSymA megaplasmid. All 29 T4cP-related genes found in GR4 have their corresponding orthologues in the reference strain Rm1021, with sequence identities of more than 98.4%. Therefore, for simplicity, in the text, we will refer to the different T4cP genes using the nomenclature of Rm1021 (https://iant.toulouse.inra.fr/bacteria/annotation/cgi/rhime.cgi accessed on 5 June 2023). In the two *S. meliloti* strains, GR4 and Rm1021, the majority of the T4cP genes are organized in two clusters: *flp-1*, located on the chromosome, and *flp-2*, on pSymA (Figure 1b). Additional orphan loci could be identified upstream of *flp-1* on the chromosome, and a third locus carrying three T4cP genes is located on pSymA (Figure 1b).

The *flp-1* cluster shows a synteny very similar to that found in *C. vibrioides* and carries all the genes needed to build a complete and functional T4cP apparatus (Figure 1 and Appendix A and Table 1). Indeed, the formation of bundle-forming pili has been associated with this chromosomal cluster in strain Rm1021 [56]. Two copies of the insertion sequence IS*Rm22* are present in the chromosomal *flp* cluster of Rm1021 [56], but in GR4, only one copy of this IS element exists (Figure 1b). Nine of the 13 genes in the *flp-1* cluster are organized within one operon [59], namely, *pilA1*, coding for the major Flp pilin; *cpaA1*, which codes for the putative prepilin peptidase; *cpaB1*, *cpaC1*, and *cpaD1*, coding for components of the outer-membrane secretin complex; *cpaE1*, coding for a putative docking protein of the secretion system; *cpaF1*, encoding the pilus assembly ATPase; and *smc02821* and *smc02822*, orthologues of the *cpaG* and *cpaH* genes [60] that code for inner-membrane platform proteins.
Figure 1Type IVc pili genes from different Gram-negative bacteria. (**a**) The *tad*/*cpa/ctp* loci in *Aggregatibacter actinomycetemcomitans* CU1000N, with gene names according to Kachlany et al. (2001) [37]; in *Caulobacter vibrioides* NA1000, with gene names according to Skerker and Shapiro (2000) [40], Christen et al. (2016) [60], and Mignolet et al. (2018) [61]; and in *Agrobacterium fabrum* C58, with gene names according to Wang et al., 2014 [41]. (**b**) Rhizobial Type IVc pili gene clusters and loci of *S. meliloti* GR4, *Sinorhizobium fredii* NGR234 [54], and *Bradyrhizobium diazoefficiens* USDA 110, with gene names according to Mongiardini et al. (2016) [53]. In the absence of assigned names, gene identifiers are used. In the case of *S. meliloti* GR4, assigned names and gene identifiers for *S. meliloti* Rm1021 were used for a short word length. Similar colors indicate similar gene products by identity or assigned function. Dark gray arrows indicate the hypothetical proteins putatively associated with pili assembly, and the light gray arrows indicate hypothetical proteins or the proteins annotated with non-pili related functions. The asterisk indicates the position of an additional IS*Rm22* element in the genome of *S. meliloti* Rm1021. ‡ BRC ID (https://www.bv-brc.org/ accessed on 4 September 2023).
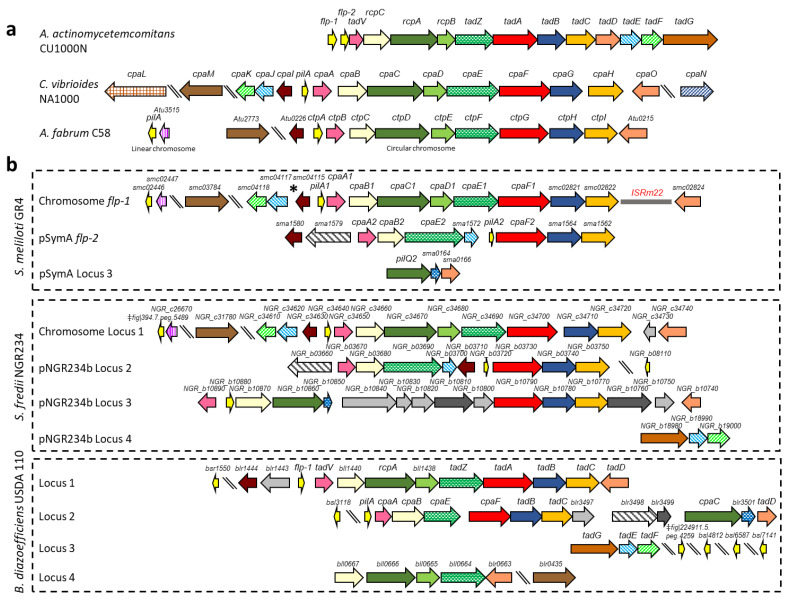



**Table 1 plants-13-00628-t001:** Percent amino acid sequence identities (in parenthesis) between the products of T4cP genes of *S. meliloti* GR4 and the closest orthologues of the *tad* genes of *A. actinomycetemcomitans* CU1000N (Aa), the *cpa* genes of *C. vibrioides* NA1000 (Cv), the *ctp* genes of *A. fabrum* C58 (Af), the *tad*/*cpa* genes of *B. diazoefficiens* USDA 110 (Bd), and the *tad*/*cpa* genes of *S. fredii* NGR234 (Sf).

Locus GR4/Rm1021	Description	Aa	Cv	Af	Bd	Sf
**Chromosome *flp-1***						
C770_GR4Chr0141/*smc04118*	CpaK-like pseudopilin		CpaK (25)		TadF (26)	NGR_c34610 (60)
C770_GR4Chr0142/*smc04117*	CpaJ-like pseudopilin		CpaJ (32)		TadE (29)	NGR_c34620 (69)
C770_GR4Chr0143/*smc04115*	CpaI-like putative pili-related protein		CpaI (26)	Atu0226 (63)	Blr1444 (35)	NGR_c34630 (88)
C770_GR4Chr0144/*smc04114* (*pilA1*)	Flp pilin subunit	Flp-2 (34)	PilA (51)	CtpA (76)	Bsr1550 (51)	NGR_c34640 (71)
C770_GR4Chr0145/*smc04113* (*cpaA1*)	Putative prepilin peptidase	TadV (31% in 65aa stretch)	CpaA (37)	CtpB (52)	TadV (43)	NGR_c34650 (71)
C770_GR4Chr0146/*smc04112* (*cpaB1*)	Component of the outer-membrane secretin complex		CpaB (35)	CtpC (64)	Bll1440 (39)	NGR_c34660 (87)
C770_GR4Chr0147/*smc04111* * (*cpaC1*)	Component of the outer-membrane secretin complex	RcpA (26)	CpaC (34)	CtpD (70)	RcpA (40)	NGR_c34670 (87)
C770_GR4Chr0148/*smc04110* (*cpaD1*)	Component of the outer-membrane secretin complex		CpaD (39)	CtpE (58)	Bll1438 (35)	NGR_c34680 (80)
C770_GR4Chr0149/*smc04109* (*cpaE1*)	Putative docking protein of the secretion system	TadZ (23)	CpaE (47)	CtpF (71)	TadZ (49)	NGR_c34690 (87)
C770_GR4Chr0150/*smc02820* (*cpaF1*)	Pilus assembly ATPase	TadA (50)	CpaF (72)	CtpG (81)	TadA (72)	NGR_c34700 (90)
C770_GR4Chr0151/*smc02821*	CpaG-like putative inner-membrane platform protein	TadB (22)	CpaG (42)	CtpH (64)	TadB Locus 1 (48)	NGR_c34710 (89)
C770_GR4Chr0152/*smc02822*	CpaH-like putative inner-membrane platform protein	TadC (21)	CpaH (48)	CtpI (69)	TadC Locus 1 (54)	NGR_c34720 (90)
C770_GR4Chr0153/*smc04116* (TRm22)	Insertion sequence IS*Rm22*					
C770_GR4Chr0154/*smc02824*	CpaO-like TPR-containing protein		CpaO (37)	Atu0215 (66)	TadD Locus 1 (45)	NGR_c34740 (84)
**Chromosomal orphan loci**						
C770_GR4Chr2732/*smc02446*	Flp pilin subunit	Flp-2 (35)	PilA (60)	CtpA (51)	Bsl4812 (51)	‡ fig|394.7.peg.5489 (60)
C770_GR4Chr2733/*smc02447*	Hypothetical protein			Atu3515 (52)		NGR_c26670 (95)
C770_GR4Chr3249/*smc03784*	CpaM-like polysaccharide deacetylase		CpaM (39)	Atu2773 (59)	Blr0435 (38)	NGR_c31780 (74)
**pSymA *flp-2***						
C770_GR4pC0512/*sma1580* *	CpaI-like putative pili-related protein		CpaI (32)	Atu0226 (31)	Blr1444 (37)	NGR_b03710 (59)
C770_GR4pC0513/*sma1579* *	Putative pili-related protein				Blr3498 (24)	NGR_b03660 (74)
C770_GR4pC0514/*sma1578* * (*cpaA2*)	Putative prepilin peptidase	TadV (30)	CpaA (38)	CtpB (36)	TadV (43)	NGR_b03670 (71)
C770_GR4pC0515/*sma1576* * (*cpaB2*)	Component of the outer-membrane secretin complex		CpaB (34)	CtpC (39)	Bll1440 (40)	NGR_b03680 (78)
C770_GR4pC0516/*sma1573* (*cpaE2*)	Putative docking protein of the secretion system		CpaE (43)	CtpF (40)	TadZ (40)	NGR_b03690 (76)
C770_GR4pC0517/*sma1572*	CpaJ-like pseudopilin protein		CpaJ (41)			NGR_b03700 (75)
C770_GR4pC0518/*sma1570* (*pilA2*)	Flp pilin subunit	Flp-1 (34)	PilA (53)	CtpA (55)	Flp-1 (62)	NGR_b03720 (69)
C770_GR4pC0519/*sma1568* * (*cpaF2*)	Pilus assembly ATPase	TadA (48)	CpaF (73)	CtpG (69)	TadA (79)	NGR_b03730 (87)
C770_GR4pC0520/*sma1564*	CpaG-like putative inner-membrane platform protein	TadB (28)	CpaG (43)	CtpH (41)	TadB Locus 1 (46)	NGR_b03740 (71)
C770_GR4pC0521/*sma1562*	CpaH-like putative inner-membrane platform protein	TadC (20)	CpaH (43)	CtpI (49)	TadC Locus 1 (52)	NGR_b03750 (84)
**pSymA Locus 3**						
C770_GR4pC1351/*sma0163* (*pilQ2*)	RcpA-like putative component of the outer-membrane secretin complex	RcpA (28)	CpaC (31)	CtpD (31)	CpaC (40)	NGR_b10860 (40)
C770_GR4pC1350/*sma0164*	Hypothetical lipoprotein				Blr3501 (33)	NGR_b10850 (29)
C770_GR4pC1349/*sma0166*	CpaO/TadD-like TPR-containing protein			Atu0215 (31% in the 70 aa middle stretch)	TadD Locus 2 (44)	NGR_b10740 (36)

* Sequence identity of 98.4–99.8% at the amino acid level between GR4 and Rm1021 gene products; otherwise 100% identical. ^‡^ BRC ID (https://www.bv-brc.org/ accessed on 4 September 2023).

The locus *smc02824*, located downstream of the IS*Rm22* element in GR4, codes for a tetratricopeptide repeat (TPR)-containing protein homologous to CpaO. Upstream of *pilA1*, the locus *smc04115* and the operon formed by *smc04117* and *smc04118* are found. These genes are *cpaI*, *cpaJ*, and *cpaK* homologues coding for a CpaC-related secretion pathway protein and TadE and TadF pseudopilins, respectively [60]. In addition, outside of the *flp-1* cluster, two additional chromosomal genes with putative roles in T4cP biogenesis were identified: *smc02446* and *smc03784* (Figure 1b, Table 1). The 57-amino acid protein encoded by *smc02446* is annotated as an Flp pilin component that contains the characteristic Flp motif recognized by the prepilin peptidase and is a paralogue of PilA1 (60% identity in a 71% query coverage) and the *flp-2*-encoded pilin PilA2 (64% identity in a 43% query coverage) (Appendix A). According to Schlüter et al. (2013) [59], *smc02446* is transcribed in an operon together with *smc02447*, which codes for a hypothetical transmembrane protein of unknown function. Finally, the orphan *smc03784* gene codes for a transmembrane protein that exhibits 39.34% identity with *C. vibrioides* CpaM, a polar determinant suggested to affect the biogenesis of functional pili portals [60,61]. Both proteins, CpaM and SMc03784, contain a C-terminal domain of divergent polysaccharide deacetylases involved in the removal of *N*-linked acetyl groups from cell-wall polysaccharides.

The *flp-2* cluster located on the pSymA megaplasmid lacks IS elements and homologues to four genes found in the chromosomal *flp-1* cluster (namely, *smc04118*, *cpaC1*, *cpaD1*, and *smc02824*). In addition, the 10 genes found in the *flp-2* cluster exhibit a different organization compared with the *flp-1* cluster (Figure 1b). Eight of the ten genes in the *flp-2* cluster are organized into two operons [59]. One of them (*pilA2-cpaF2-sma1564-sma1562*) comprises paralogous genes of the *flp-1* cluster that encode a Flp pilin, a pilus assembly ATPase, and the two platform proteins of the pilus apparatus, whereas the second operon (*cpaA2-cpaB2-cpaE2-sma1572*) includes paralogous genes of the *flp-1* cluster that encode a prepilin peptidase, a secretin-associated protein, a docking protein for the secretion system, and a pseudopilin. Upstream of *cpaA2*, two other genes with putative roles in pilus biogenesis could be identified: *sma1580*, encoding a CpaI homologue, and *sma1579*, coding for a membrane-associated protein with an N-terminal Tad_N domain found in a family of putative Flp pilus-assembly proteins. No genes homologous to *rcpA*/*cpaC*/*ctpD,* coding for the outer-membrane secretin, were found associated with *flp-2*, suggesting that this cluster codes for an incomplete secretion system (Appendix A). Interestingly, outside of the *flp-2* cluster on pSymA, a locus (Locus 3) of three genes (*pilQ2-sma0164-sma0166*) could be identified, which contains a homolog of *cpaC* and a gene coding for a protein with characteristic features of T4P pilotins (Figure 1b and Appendix A and Table 1).

The *pilQ2* gene (its name denotes its homology to the PilQ secretin of T4aP) of Locus 3 codes for a 466-amino acid (aa) protein with a signal-peptide sequence and a secretin-like domain (Appendix A). This protein shows 31% identity with both *C. vibrioides* CpaC and *A. fabrum* CtpD and 30% identity with its chromosomal paralogue, CpaC1 (Table 1, Appendix A). Downstream of *pilQ2* and forming part of the same operon [59] is *sma0164*, which codes for a hypothetical 98 aa protein that exhibits significant similarities with proteins encoded by *NGR_b10850* and *blr3501*, which are genes associated with the T4cP loci in *Sinorhizobium fredii* NGR234 and *B. diazoefficiens*, respectively (Figure 1b and Appendix A). The three proteins contain a prokaryotic membrane-lipoprotein lipid attachment profile as identified by InterPro (Appendix A). The profile comprises a precursor signal peptide containing the LXXC lipobox motif that is processed by signal peptidase 2, which cuts upstream of the conserved cysteine residue to which a diacylglycerol is attached [62,63]. The third gene of Locus 3 is *sma0166*, which encodes a conserved hypothetical protein of 176 aas. Although this protein does not exhibit significant homology with CpaO from *C. vibrioides* or TadD from *A. actinomycetemcomitans*, all three of these proteins contain a TPR motif (Appendix A). This motif, which is involved in protein–protein interactions and in the assembly of multiprotein complexes [64], is also found in the pilotins TadD and PilF of *Pseudomonas aeruginosa* [43]. Moreover, and although PROSITE does not identify a prokaryotic membrane-lipoprotein profile associated with SMa0166, a lipobox motif is found in the N-terminal region of the protein (Appendix A). The presence in Locus 3 of genes putatively coding for a CpaC homologue and a putative pilotin opens the possibility that this locus complements the *flp-2* cluster to function as a complete Flp pilus biogenesis and secretion system.

*S. meliloti* T4cP-related genes identified in the bioinformatic analyses were compared to T4cP-related genes reported in bacteria belonging to *Rhizobiaceae*, namely, the *ctp* genes of *A. fabrum* C58 [41], genes of *S. fredii* NGR234 [54], and the *tad*/*cpa* genes of *B. diazoefficiens* USDA 110 [53] (see Figure 1, Table 1). As expected, the highest similarity, both in gene organization as well as in sequence, is shared between gene clusters from the same genera, *Sinorhizobium*. In this manner, the *flp-1* gene cluster of *S. meliloti* shares the highest similarity with the chromosomal Locus 1 of *S. fredii* NGR324 while the *flp-2* cluster has the highest similarity with Locus 2, located on the pNGR234b replicon of *S. fredii* NGR234. The cluster of genes in Locus 3 of *S. meliloti* GR4 and Sm1021 share the highest similarity with the *cpaC-blr3501-tadD* genes of Locus 2 of *B. diazoefficiens*. The operon encoding the additional Flp pilin found in the chromosome outside of the *flp-1* cluster is most similar to that found upstream of Locus 1 in *S. fredii* NGR234 and in the linear chromosome of *A. fabrum* C58. In general, the sequence identity between the gene products of the *tad*/*cpa* gene clusters is higher with orthologues in other bacteria than with paralogues of *S. meliloti* GR4 (Table 1 and Appendix A), suggesting that their origin is not due to gene duplication events within the same strain.

### 2.2. Both flp-1 and flp-2 Clusters Contribute to T4cP Production in S. meliloti GR4

In strain Rm1021, the formation of Flp pili has been associated with the *flp-1* cluster since pilus-like filaments were mostly absent in cells with a *pilA1* deletion [56]. However, the rare observation of pilus-like structures in a small percentage of *pilA1* mutant cells led the authors to suggest the possibility of pili formation associated either with the expression of the *flp-2* cluster or the incorporation of the Flp pilin encoded by *smc02446*. To confirm whether both *flp-1* and *flp-2* clusters are responsible for the formation of Flp pili in *S. meliloti* GR4, single and double partial-cluster mutants were constructed. Although this approach has the drawback of making complementation experiments more difficult, it has the advantage of reducing the probability of obtaining inconclusive results due to the possible complementary effect of paralogous genes. In the GRflp1 mutant, the *pilA1*, *cpaA1*, *cpaB1*, *cpaC1*, *cpaD1*, *cpaE1*, and *cpaF1* genes were deleted, whereas the GRflp2 mutant harbored a deleted version of the *flp-2* cluster in which the *pilA2*, *cpaF2*, *sma1564*, and *sma1562* genes had been removed (Figure 2). In addition, the GRflp1flp2 double mutant was generated with deleted forms of both the *flp-1* and *flp-2* clusters.

The wild-type and *flp* single- and double-mutant strains were examined for the presence of Flp pili by transmission electron microscopy (TEM). For this purpose, cells grown on solid media were used. Two different cell-surface appendages were observed in the wild-type *S. meliloti* GR4: flagella corresponding to the thicker and more curved filaments, and less abundant, thinner, and straighter pili-like structures that were often found forming bundles and connecting bacterial cells (Figure 3a,b and Appendix A). The characteristics shown by the thinner filaments resembled those described for Flp pili of different bacteria, including *S. meliloti* Rm1021 [41,48,56]. However, neither in the single *flp* mutants nor in the double mutant could pili-like structures be observed, and only flagella were evident as cell-surface appendages (Appendix A). In order to facilitate the detection of pili structures, *flp* mutants were created in the flagellaless background offered by the *flaAflaB* mutant. Due to the lack of the major flagellin, FlaA, the *flaAflaB* mutant cannot produce normal functional flagella; instead, short straight filamentous structures resulting from mutant broken filaments can be observed [10]. Micrographs of the different flagellaless strains revealed that, in addition to the short broken flagellar filaments, pili-like structures were also evident in the *flaAflaB* mutant (Figure 3c,d) and its *flp* simple derivative mutants (Figure 3e–h) but not in the flagellaless *Δflp-1Δflp-2* double mutant (Figure 3i,j). This suggests that both *flp* clusters function to make pili-like structures.

### 2.3. Type IVc Pili (T4cP) Facilitate the Surface Spreading of S. meliloti GR4

Pili can mediate different types of surface motility [32]. In *C. crescentus*, Tad pili are able to promote twitching- or walking-like movements [65]. To evaluate the effect of T4cP in *S. meliloti* GR4 translocation, wild-type as well as single and double *flp*-mutant-derivative strains were assessed in different motility assays. Using different media, agar sources, and concentrations under the experimental conditions known to promote twitching motility in different bacteria [66], similar limited spreading in the agar/plate interphase was observed for all of the strains (Appendix A). Since no differences were detected between the wild-type and the *flp* mutants in this type of assay, it remains unclear whether the limited spreading observed was due to twitching or to other forms of bacterial motility. Nonetheless, it appears that the *flp* clusters are not involved in this trait. In contrast, when motility was assessed under conditions that promote swarming in strain GR4 [10], some differences were observed between the wild-type and *flp*-derivative mutants. Of the 18 replicates, the frequency of swarming observed for the parental strain was 100%, while 89%, 61%, and 67% frequencies were observed for the deletion mutants *Δflp-1*, *Δflp-2*, and *Δflp-1Δflp-2*, respectively. Pairwise comparison using the Chi-square statistic indicated a significant difference (*p* < 0.01) between the swarming frequency of the wild-type compared to either *Δflp-2* or *Δflp-1Δflp-2* but not *Δflp-1*. With respect to the magnitude of surface spreading, surface motility was affected in all mutant strains and significantly so in the *Δflp-1Δflp-2* double mutant (Figure 4a,b). These results suggest that the inactivation of the *flp-2* cluster has a stronger effect on surface motility than the inactivation of the *flp-1* cluster, and that this effect is cumulative in the double mutant. When swimming motility was tested in BM (0.3%), all three *flp* deletion mutants exhibited a slight but significant decrease in their swimming halo compared with that of the wild-type strain, with both the single mutants and the double mutant being equally affected (Figure 4c,d).

### 2.4. The Interaction of S. meliloti GR4 with Alfalfa Is Influenced by T4cP

Deletion of the *pilA1* gene in *S. meliloti* strain Rm1021 reduced the competitiveness for nodule occupancy when confronted with the wild-type, indicating that Flp pili play a role in Rm1021’s symbiotic performance [56]. However, to the best of our knowledge, the role of the pSymA-encoded Flp pili in symbiosis has not yet been reported. In addition, *S. meliloti* strains Rm1021 and GR4 present remarkable differences, including some symbiotic properties, with the latter being more efficient in colonizing alfalfa roots and exhibiting a higher competitive ability than Rm1021 [9,13]. In this study, the contribution of chromosomal- and pSymA-encoded T4cP to the interaction of GR4 with alfalfa plants was investigated. No differences in bacterial attachment to alfalfa roots were found between the wild-type strain or any of the single *flp* mutants as indicated by the similar number of colony-forming units (CFUs) associated with plant roots 2 h after inoculation (Figure 5). Likewise, the single *flp* deletion mutants colonized plant roots as efficiently as the wild-type after 24, 48, and 72 h (Figure 5). In contrast, we found that the double Δ*flp-1*Δ*flp-2* mutant shows increased adhesion to root surfaces 2 h post-inoculation compared with the wild-type and the single *flp* mutants (Figure 5). Remarkably, the increased capacity to attach to root surfaces exhibited by the double mutant does not result in increased bacterial colonization of the roots. Indeed, 72 h after inoculation, the number of Δ*flp-1*Δ*flp-2* CFUs associated with roots was lower than in the wild-type, although this difference was not significant. These results indicate that the presence of either of the two Flp pili produced by *S. meliloti* reduces bacterial attachment to alfalfa roots, but this effect does not preclude efficient colonization of the plant host.

Next, to determine whether the *flp* clusters had an effect on the symbiotic interaction with alfalfa plants, the infectivity and competitive abilities of the single and double *flp* mutants were assayed (Figure 6). All three mutants were able to induce the formation of nitrogen-fixing nodules, and no differences in the nodule formation efficiency or alfalfa growth were detected in the single *flp* mutants compared with the wild-type (Figure 6a and Appendix A). However, the number of nodules induced by the double mutant was slightly but significantly lower than those of the wild-type or single *flp* mutants at the end of the experiment (13 and 15 days post-inoculation) (Figure 6a). We also determined the competitive ability for nodule occupancy of the *flp* mutants by inoculating alfalfa plants with bacterial mixtures of the strain to be tested and a marked wild-type strain (*S. meliloti* GR4 (pGUS3)) at a ratio of 1:1. These competition assays showed that the *Δflp-2* mutant showed a significantly lower nodule occupancy than the wild-type strain (Figure 6b). Notably, this reduction in competitiveness was not observed in the double *Δflp-1Δflp-2* mutant (Figure 6b), indicating that the lack of T4cP from cluster *flp-2* reduces the bacterial competitive ability for nodule occupation only when T4cP from the *flp-1* cluster are present. Altogether, these results demonstrate that both *flp-1*- and *flp-2*-dependent pili influence the *S. meliloti* symbiotic fitness.

## 3. Discussion

This work provides insights into T4cP in rhizobia by uncovering the role of the two *flp* clusters present in the highly competitive *S. meliloti* strain, GR4. We show that both *flp-1* and *flp-2* are functional in the biogenesis of pili-like structures, and both contribute to the optimal surface translocation exhibited by this rhizobial strain. Our data indicate that although these *flp* clusters are dispensable for the formation of nitrogen-fixing nodules, they modulate symbiotic characteristics that might confer adaptive advantages such as bacterial attachment to plant roots, nodule formation efficiency, and bacterial competitiveness for nodule occupation.

Despite the participation of T4cP in diverse microbial lifestyles [30,32,47], it is striking that, to date, only two studies have addressed the role of T4cP genes in two different rhizobial species [53,56]. Most likely, the abundance of *tad/cpa* genes in rhizobial genomes [11,38], together with the subtle phenotypes exhibited by T4cP-related mutants under laboratory conditions, discouraged further investigation of these filaments in legume symbionts. Our bioinformatic analyses identified up to 29 genes putatively related to T4cP in *S. meliloti*. This number contrasts with the relatively lower number of *tad*/*cpa* genes (14–16 genes) found in *A. actinomycetemcomitans* and *C. vibrioides* [35,40,60]. Remarkably, the number of loci putatively related to T4cP is even higher in other rhizobia such as *S. fredii* NGR234 (47 genes) and *B. diazoefficiens* USDA 110 (39 genes). The maintenance of such a number of pili-related genes throughout evolution could be an indication of the significant adaptive advantages conferred by pili to rhizobial lifestyles. Another remarkable feature of T4cP-related genes in rhizobia compared with *tad*/*cpa* genes in *A. actinomycetemcomitans* and *C. vibrioides* is their organization into various clusters. In addition to a chromosomal cluster that appears to be conserved in phylogenetically distant bacteria, the three rhizobial species analyzed in our study contain other clusters located in either the chromosome (*B. diazoefficiens*) or in megaplasmids (pSymA in *S. meliloti* and pNGR234b in *S. fredii* NGR234). The existence of more than one cluster of *tad*/*cpa* genes is not unique to rhizobia [38,48,67]. Like in *S. meliloti*, in the bacterial wilt pathogen *Ralstonia solanacearum*, two distinct *tad* clusters located in the megaplasmid and chromosome were identified [48]. Characterization of a *tadA2* mutant of this bacterium revealed that the Flp pili associated with the megaplasmid cluster are required for virulence in potato plants [48], whereas no information exists regarding the function of the chromosomal cluster. The opportunistic animal pathogen *Vibrio vulnificus* harbors three *tad* loci, but only the triple mutant exhibited decreased virulence in mice, indicating that the *tad* loci work cooperatively in this species during pathogenicity [67].

In *S. meliloti,* the chromosomal *flp-1* cluster carries all of the genes needed for T4cP biogenesis, and the formation of bundle-forming pili was previously associated with this cluster in strain Rm1021 [56]. In contrast, *flp-2* lacks some key genes for the secretion system to be functional. The lack of *cpaK* and *cpaD* homologues should not be an impediment for T4cP biogenesis. Indeed, *A. fabrum* and *Pseudomonas aeruginosa* are capable of assembling Flp pili in the absence of *cpaK* and *cpaD* homologues, respectively [41,68]. However, the lack of genes coding for components of the outer-membrane secretin channel (*cpaC* and *cpaO*) could prevent pilus biogenesis associated with the *flp-2* cluster. Interestingly, an additional set of three genes located on pSymA (Locus 3) was identified that could complement the genes in the *flp-2* cluster. Indeed, TEM analysis of flagellaless mutants carrying single large deletions in either the *flp-1* or *flp-2* clusters revealed the production of filaments, while these fimbriae structures were no longer observed in a flagellaless double *flp-1flp-2* mutant. These results indicate that, under our experimental conditions, the two *flp* clusters are functional in the formation of pili. In the study carried out by Zatakia and co-workers (2014) [56], pili-like structures were only rarely observed in the *pilA1* mutant of Rm1021, which the authors explained as the result of T4cP assembly from the *flp-2* cluster and/or the incorporation of the pilin encoded by the orphan *smc02446* gene. The latter is a very likely possibility considering that the two pilin genes in *A. fabrum*, *ctpA* and *pilA*, are functionally interchangeable [41]. The approach used in our study, in which deletions of gene clusters were generated in the *flp* regions instead of deleting individual genes, reduces the possibility of inconclusive results due to possible complementation effects of paralogous genes. In our *flaAflaB*Δ*flp-1* mutant, in which not only *pilA1* but also *cpaA1, cpaB1, cpaC1, cpaD1, cpaE1*, and *cpaF1* were deleted, pili-forming bundles connecting cells were clearly observed that cannot be the result of the incorporation of the *smc02446*-encoded pilin. Moreover, pili-like structures could not be detected in *flaAflaB*Δ*flp-1*Δ*flp-2*, indicating that the filaments observed in *flaAflaB*Δ*flp-1* are due to a fully functional *flp*-2 cluster, possibly with the implication of Locus 3, which contains all the three genes missing in *flp-2*. However, experiments are still needed to demonstrate that Locus 3 is essential for pili biogenesis associated with the *flp-2* cluster.

The splitting of genes required for the assembly of T4cP in distinct genetic clusters is not usual [31], but at least one example has been reported in the literature. In the bacterial predator *Myxococcus xanthus*, the assembly of the Tad-like Kil system involved in contact-dependent prey killing is the result of the expression of two clusters that carry complementary sets of genes [47]. In addition, our bioinformatic analyses reveal that in *S. fredii*, Locus 2 on pNGR234b shows high similarity, both in gene organization and sequence, with *S. meliloti flp*-2 and, like the latter, it also lacks genes coding for the secretin complex. Homologues to *S. meliloti* Locus 3 genes can be identified in a different region in pNGR234b (*NGR_b10860*, *NGR_b10850*, and *NGR_b10740* in Locus 3), albeit exhibiting a different gene organization and clustered together with additional T4cP-related genes. Clearly, further investigations are needed in order to elucidate if distinct co-existing *tad/flp* clusters in rhizobia play different or specific roles or if they have complementary functions.

In this work, pili-like structures were more easily detectable in the flagellaless strains containing one or two *flp* clusters than in the wild-type GR4 strain (Figure 3). A simple explanation could be that the abundant flagellar filaments present in the wild-type strain are hampering the observation of the thinner and scarcer pili-like structures. However, the possibility that the lack of flagellar filaments has an impact on T4cP gene expression in *S. meliloti* should not be discarded and deserves further investigation. In bacteria in which T4cP have been studied the most, expression of the pilus-associated genes and activity of the filament are controlled by complex transcriptional and posttranslational regulatory mechanisms [32,38]. In *P. aeruginosa*, a feedback regulation between pili and flagellar components occurs through a two-component regulatory system [69]. Moreover, *C. crescentus* flagellar mutants are significantly deficient in pili biogenesis, and it has been suggested that different stages of the flagellum assembly act as checkpoints for the regulation of T4cP-associated gene expression [70]. Another open question that was not addressed in this study is whether the two *flp* clusters in *S. meliloti* are expressed under the same conditions. The presence of an accessory chemotaxis system next to the *flp-2* cluster in pSymA is intriguing. This chemotaxis system belongs to the alternative cellular function (ACF) and most likely does not control flagella [11]. It would be worth investigating whether this chemotaxis system controls the activity of T4cP in *S. meliloti* in an analogous way to how the Pil-Chp pathway controls T4aP in other bacteria [71].

Tad pili are able to promote twitching- or walking-like movements in *C. crescentus* [65]. Surface-motility assays performed with our *S. meliloti flp* deletion mutants indicate that the *flp* clusters contribute cooperatively to the optimal surface motility exhibited by strain GR4, with the *flp-2* cluster having a stronger effect than *flp-1*. Under the experimental conditions used in our study, surface motility in GR4 is totally dependent on flagellar action [10]. At present, we can only speculate about how *flp-1* and *flp-2* affect swarming motility in GR4. Our data indicate that the double *flp-1flp-2* mutant exhibits greater attachment to alfalfa roots compared with the wild-type (Figure 5). An increased adhesion to surfaces facilitated by the absence of pili could hamper proper bacterial translocation. On the other hand, pili together with flagella participate in surface sensing in bacteria [72], and, specifically, Tad pili act as surface contact sensors for *C. crescentus* [33,65,73]. An attractive possibility awaiting investigation is that pilus-based mechanosensation in *S. meliloti* triggers a signal transduction cascade that promotes swarming and other physiological adaptations to thrive on surfaces.

Information about the role of T4cP in the establishment of symbiosis with legumes is very scarce. Results obtained in this study indicate that T4cP impact the symbiotic fitness of *S. meliloti* by affecting adhesion to plant roots, nodule formation efficiency, and competitiveness. The role of rhizobial T4cP in favoring adhesion to plant roots has been previously proposed in two studies [53,56]. In *B. diazoefficiens* USDA 110, inactivation of *tadG* or the cluster *tadGEF* (Locus 3 of *B. diazoefficiens* USDA 110 in Figure 1b) impaired adhesion to soybean roots and caused delayed root infection [53]. However, the connection between the symbiotic phenotype of *tadGEF* mutants and pili formation in *B. diazoefficiens* USDA 110 was not directly examined. In contrast to *B. diazoefficiens* and *S. fredii* NGR234, no *tadG* homologue could be identified in the genome of *S. meliloti*. On the other hand, the reduced competitive ability for nodule occupation exhibited by a *S. meliloti pilA1* mutant was suggested to be due to a lower bacterial attachment to plant roots, although this possibility was not experimentally tested [56]. Results obtained in our study do not support a role for *S. meliloti* T4cP in promoting adhesion to root surfaces. As already discussed, the double *flp-1flp-2* mutant, which lacks pili-like structures, exhibits greater attachment to plant roots compared with the wild-type and single *flp* mutant strains (Figure 5). These observations demonstrate that T4cP are not required for the attachment of *S. meliloti* cells to alfalfa roots and suggest that the presence of T4cP hamper attachment to the root surface mediated by other cell surface structures, perhaps flagella, exopolysaccharides, and/or lipopolysaccharides [4].

Our data also demonstrate that the increased binding to root surfaces shown by the double *flp*-cluster mutant does not significantly affect the biofilm formed on plant roots at 24, 48, or 72 h after inoculation, indicating that, under our experimental conditions, *S. meliloti* T4cP are not essential for plant root colonization during the early stages of the interaction. Likewise, in *A. fabrum*, no differences were detected between the 48 h biofilm formed on *Arabidopsis thaliana* roots by the wild-type strain and a *ctpA*-derivative mutant [41]. Nevertheless, our *S. meliloti* double *flp* mutant showed decreased nodule formation efficiency, as indicated by the significantly lower number of nodules induced by this strain compared with the wild-type or single *flp* mutants. At present, the reasons for the lower nodulation performance of the double *flp* mutant are unknown. Perhaps the increased attachment to the root surface shown by the mutant together with the lower ability to move across surfaces reduces the probability of rhizobial cells to find potential infection sites in a timely manner. *S. meliloti* invades legume roots through plant-derived infection threads that initiate only on growing root hairs located near the root tip [1,74]. Since root hairs remain infectable for only a few hours, the rhizobial ability to detach and move across the root surface to find new emerging root hairs might positively impact the nodule-formation efficiency. Another, non-excluding possibility is that T4cP, directly or indirectly, contribute to defending *S. meliloti* cells against plant defense responses in a role similar to the stealth-hiding role proposed for the Tad pili in *V. vulnificus* [67].

Finally, our data show that deletion of *flp-1* in GR4 does not affect its competitive ability for nodule occupation. This result contrasts with the lower competitiveness exhibited by a *pilA1* deletion mutant compared with its corresponding wild-type strain, Rm1021 [56]. Differences between GR4 and Rm1021 and/or the type of mutation used to analyze the role of T4cP associated with *flp-1* could explain the contrasting results obtained in the two studies. Interestingly, although the lack of *flp-1* has no influence on GR4 competitiveness, the presence of this cluster in a *flp-2* deletion mutant reduces its ability to compete for nodule occupation. This is based on the lower nodule occupancy exhibited by the single *flp-2* mutant that is recovered when *flp-1* is deleted. A possible explanation for this result is that in the absence of Flp-2 pili, interbacterial attachment mediated by Flp-1 pili is increased, which could hamper dissemination to other infection sites. This seems not to be crucial in single inoculation experiments, as shown by similar nodulation kinetics of the *flp-2* mutant and the wild-type strain but confers a disadvantage when competing with another bacterium.

In summary, this study increases our knowledge of T4cP in *S. meliloti* and their role in the interaction with its plant host and, at the same time, highlights the complexity of their study. New questions have arisen, and investigations aimed at answering them might help us to better understand how extracellular filament appendages contribute to the symbiotic fitness of rhizobia.

## 4. Materials and Methods

### 4.1. Bacterial Strains, Plasmids, and Growth Conditions

Bacterial strains and plasmids used in this work are listed in Appendix A. *Escherichia coli* cultures were routinely grown in LB medium [75] at 37 °C. *S. meliloti* strains were grown in either complex tryptone yeast (TY) medium [76] or in Robertsen minimal medium (MM) [77] at 28 °C. When required, antibiotics were added to final concentrations (µg mL^−1^) of 50 kanamycin and 25 hygromycin for *E. coli*; 200 kanamycin and 75 hygromycin for *S. meliloti*. All reagents were obtained from Sigma-Aldrich (Steinheim, Germany), unless otherwise specified.

### 4.2. In Silico Analyses

In order to identify and compare *tad/cpa/ctp* genes from *S. meliloti* GR4, *A. actinomycetemcomitans* CU1000N, *C. vibrioides* NA1000, *A. fabrum* C58, *S. fredii* NGR234, and *B. diazoefficiens* USDA110, genome sequences were retrieved from the NCBI database (https://www.ncbi.nlm.nih.gov/datasets/genome/ accessed on 4 September 2023). Comparisons of these genomes were performed in the Rapid Annotations mode using the Subsystems Technology (RAST) server version 2.0 [78]. Further gene identification was performed using resources at KEGG (https://www.genome.jp/kegg/ accessed on 4 September 2023 [79]) and BV-BRC (https://www.bv-brc.org/ accessed on 4 September 2023 [80]). Sequence comparisons of individual gene products were performed with blastp (https://blast.ncbi.nlm.nih.gov/Blast.cgi accessed on 4 September 2023) and Clustal MUSCLE sequence alignment at https://www.ebi.ac.uk/Tools/msa/muscle/ (accessed on 4 September 2023). Protein domains within individual gene products were examined using InterPro (https://www.ebi.ac.uk/interpro/ accessed on 4 September 2023 [81]).

### 4.3. Construction of S. meliloti Mutant Strains

*flp-1* and *flp-2* mutants were obtained by the allelic exchange of each *flp* cluster with markerless deleted versions obtained in vitro by overlap–extension PCR using primers delpilA1.1 to delpilA1.4 and delpilA2.1 to delpilA2.4 (Appendix A) and the Phusion high-fidelity DNA polymerase. The resulting amplicons corresponding to deleted versions of *flp-1* and *flp-2* were first cloned into pCR-XL-TOPO and confirmed by DNA sequencing and then subcloned into pK18*mobsacB* as *Bam*HI or *Hin*dIII fragments to produce pK18-Δflp1 and pK18-Δflp2, respectively. Each one of these constructions was introduced into *S. meliloti* GR4 by conjugation using S17-1 as the donor strain [82], and the allelic exchange was selected as previously described [83] to obtain the single GRflp1 and GRflp2 mutants. The double GRflp1flp2 mutant was generated by an allelic exchange of the *flp-2* cluster with the corresponding deleted version in the single GRflp1 mutant. To obtain flagellaless *flp* mutants, plasmid pK18flaAB::Hyg was introduced into the single and double *flp* mutants, and the replacement of wild-type flagellin genes with the mutated version was selected for on plates containing hygromycin. All mutants were checked by Southern hybridization with specific probes.

### 4.4. Transmission Electron Microscopy (TEM)

Cells for TEM observations were obtained from the edge of colonies grown on solid MM (1% agar) using the same procedures as that described in Calatrava-Morales et al. (2017) [84]. In brief, carbon-coated Formvar grids were placed for 5 min on top of a drop of water (sterile MilliQ) previously applied to the colony border. The grids were then washed twice in MilliQ water for 1 min and stained with 2% (wt/vol) uranyl acetate for 3 min. Grids were allowed to air dry for at least 1 h and were visualized using a JEOL (Tokyo, Japan) JEM-1011 transmission electron microscope with a 100 kV beam at the Microscopy Service of the Estación Experimental del Zaidín (Granada, Spain). Images were captured with an Olympus (Münster, Germany) SIS Megaview III CCD camera for TEM.

### 4.5. Motility Assays

To determine swimming motility, *S. meliloti* strains were grown in liquid TY medium at 28 °C until they reached an optical density (OD_600nm_) of approximately 0.5. Aliquots of 3 μL of each culture were placed onto plates containing Bromfield medium (BM) (0.04% tryptone, 0.01% yeast extract, 0.01% CaCl_2_ × 2H_2_O, 0.3% Bacto Agar, Difco BM). After incubation at 28 °C for 96 h, the diameter of the dispersion halo of each colony was measured.

Swarming-motility assays were performed as described by López-Lara et al. (2018) [85]. Briefly, cultures of the *S. meliloti* strains in liquid TY medium were grown overnight to an OD_600_ of greater than 1. Subsequently, each culture was washed twice with liquid MM, and after the last wash, the pellet was resuspended in 80 μL of liquid MM. Aliquots of 2 μL of the suspension were inoculated and allowed to dry for 10 min on the surface of MM plates (0.6% noble agar, Difco BM), which had been previously dried for 15 min. The plates were incubated at 28 °C for 48 h. To quantify motility, the average length of the two sides of a rectangle able to exactly frame each colony was measured in millimeters. Colonies that had extended 7 mm or further were considered positive in swarming.

Twitching motility was assayed according to the protocol described by Turnbull and Whitchurch (2014) [66], with small modifications. Plates containing 10 mL of either TY (1% Bacteriologic Agar), MM (1.3% Agar, Grade A Difco), MM (1% Agar, Grade A Difco), or MM (1% Agar, Noble) were prepared and dried according to the protocol. Strains were stab-inoculated with a toothpick to the bottom of the plate and incubated for 2–4 days at 28 °C in a humidified chamber.

### 4.6. Plant Assays

Alfalfa (*Medicago sativa* L. ‘Victoria’) plants were surface sterilized, germinated, and grown in hydroponic cultures under axenic conditions in glass tubes containing nitrogen-free nutrient solution, as previously described [86].

To test the degree of infectivity of each strain, 9-day-old plants (18–20 replicates, one plant per tube) were inoculated with 1 mL of a rhizobial suspension containing 5 × 10^6^ cells. Prior to inoculation, bacteria were grown to the exponential phase (OD_600_ = 0.5 to 0.6) in TY broth and diluted 100-fold in sterile water. Four days after inoculation, the number of nodules per plant was recorded over two weeks.

To examine competitive ability, 12 individual plants grown as described for the infectivity test were inoculated with 1 mL of a bacterial suspension prepared as already described, containing mixtures at a ratio of 1:1 of the *S. meliloti* strain to be tested and GR4 marked with the pGUS3 plasmid [87]. Roots were collected 15 days after inoculation, briefly washed with water, and incubated overnight in the dark at 37 °C in 1 mM 5-bromo-4-chloro-3-indolyl-b-D-glucuronide (Apollo Scientific, Cheshire, UK) in 50 mM sodium phosphate buffer (pH 7.5) with 1% SDS. Nodules containing the strain harboring the pGUS3 plasmid, either as the sole occupant or together with the unmarked strain (double occupancy), stain blue. In contrast, nodules exclusively occupied by the unmarked strain remain white. Nodule occupancy was determined by counting blue and white nodules in three independent experiments and by determining the percentage of white nodules obtained for each inoculation mixture. Significant differences with respect to the percentage of white nodules obtained in the control inoculation mixture (GR4 vs. GR4pGUS3) indicate differences in competitiveness between GR4 and the tested strain.

Colonization assays were performed as described by Calatrava-Morales et al. (2017) [84]. Briefly, 1-week-old alfalfa plants (5 plants/tube) were inoculated with 1 mL of rhizobial suspension as described above. At different times, 15 roots from each treatment were collected and washed three times with sterile distilled water, and pools of roots from 5 plants were introduced into an Eppendorf tube containing sterile Tris-EDTA buffer (10 mM Tris-HCl [pH 7.5] and 1 mM EDTA). Tubes were sonicated (2× for 1 min each) in a Selecta (Abrera, Spain) Ultrasons Sonicator bath. Released rhizobia were quantified by counting the number of colony-forming units (CFU) from serial dilutions, and data were plotted as CFU per gram of root fresh weight (RFW).

## Figures and Tables

**Figure 2 plants-13-00628-f002:**
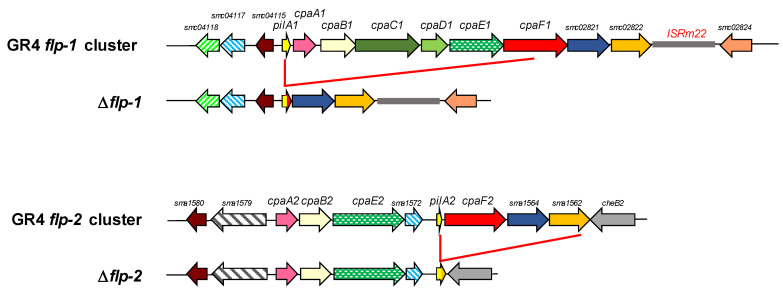
Genomic regions corresponding to the *flp-1* and *flp-2* clusters of *S. meliloti* GR4 and their corresponding unmarked in-frame deleted versions obtained in this study. In Δ*flp-1*, the *pilA1*, *cpaA1*, *cpaB1*, *cpaC1*, *cpaD1*, *cpaE1*, and *cpaF1* genes were deleted, whereas in Δ*flp-2*, the *pilA2*, *cpaF2*, *sma1564*, and *sma1562* genes were removed. Color codes are the same as those used in Figure 1.

**Figure 3 plants-13-00628-f003:**
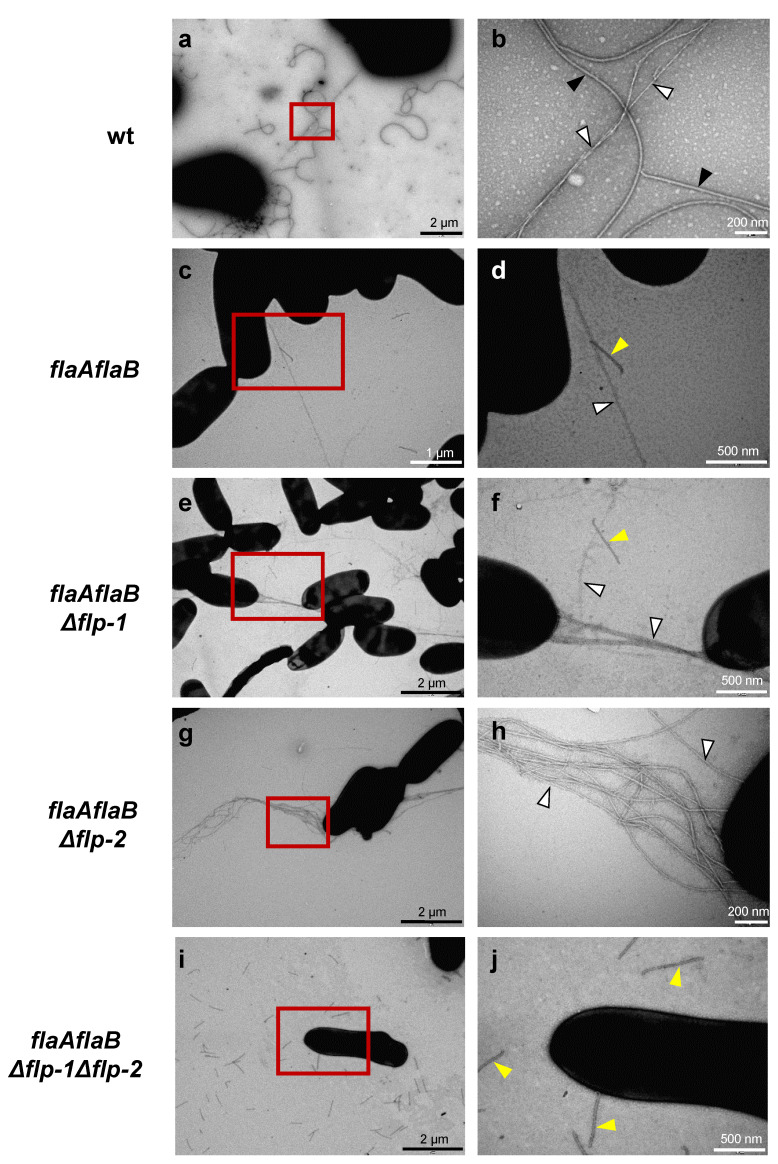
Transmission electron microscopy of *S. meliloti* strains. Micrographs were obtained from cells grown overnight on solid MM (1%) and negatively stained with uranyl acetate (2%). Each right panel is a closer view of the red box indicated in the corresponding left panel. Wild-type GR4 strain (**a**,**b**) and flagellaless strains: GR4flaAB (**c**,**d**), GRflaflp1 (**e**,**f**), GRflaflp2 (**g**,**h**), and GRflaflp1flp2 (**i**,**j**). Black and white arrowheads indicate flagella and Flp pili, respectively. Yellow arrowheads indicate the short broken flagellar filaments characteristic of *flaAflaB* mutants.

**Figure 4 plants-13-00628-f004:**
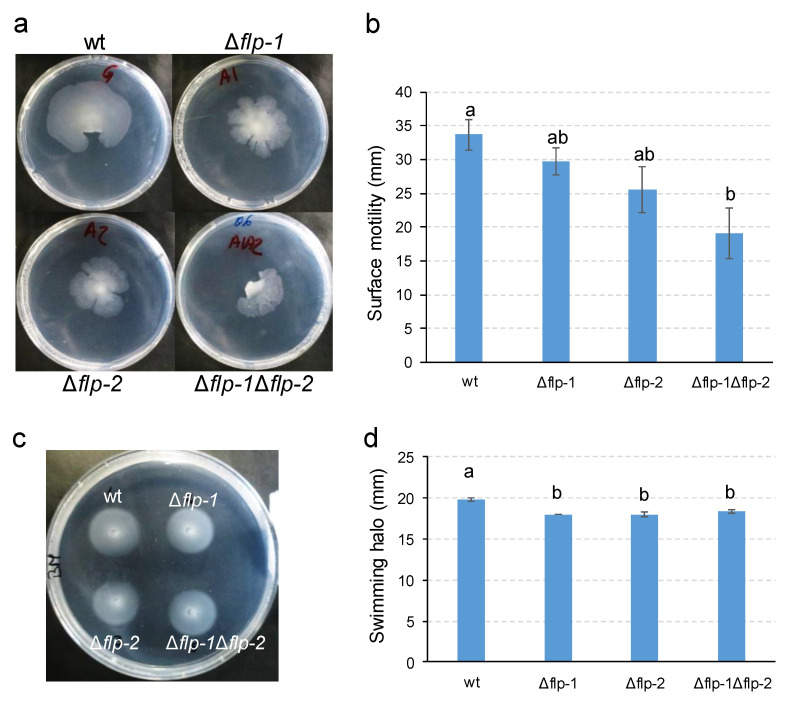
Surface and swimming motilities exhibited by GR4-derivative *flp* deletion mutants. (**a**) Representative pictures and (**b**) graph of the surface motilities exhibited after 48 h of incubation on MM (0.6% agar). Means and standard errors were obtained from at least 2 replicates in three independent experiments. (**c**) Representative picture and (**d**) graph of the swimming motilities in BM (0.3% agar) after 96 h of incubation. Means and standard errors were obtained from 3 replicates. The different letters indicate significant differences according to an analysis-of-variance (ANOVA) test (*p* ≤ 0.05).

**Figure 5 plants-13-00628-f005:**
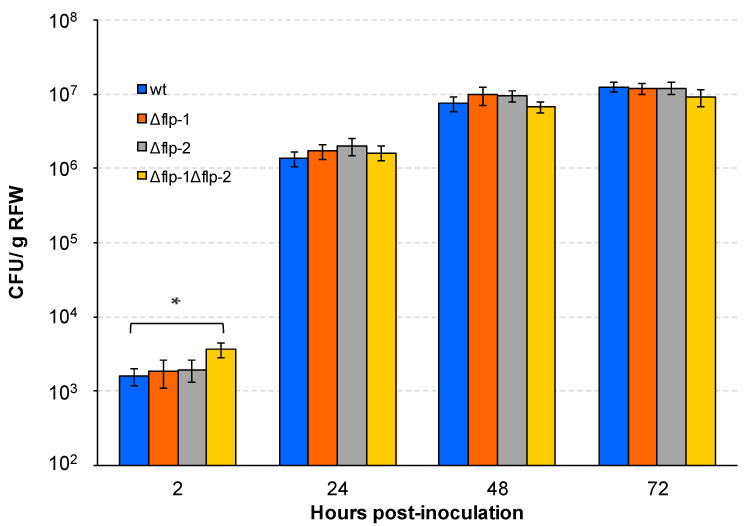
Bacterial attachment to and colonization of alfalfa roots by *flp* deletion mutants of *S. meliloti*. Colony-forming units (CFU) recovered from alfalfa roots were determined 2 h after inoculation to determine bacterial adhesion and at 24, 48, and 72 h post-inoculation to assess the colonization ability of each bacterial strain. Data are expressed per gram of root fresh weight (RFW). Error bars indicate the standard error of the mean. The asterisk indicates a significant difference according to an ANOVA test (*p* ≤ 0.05).

**Figure 6 plants-13-00628-f006:**
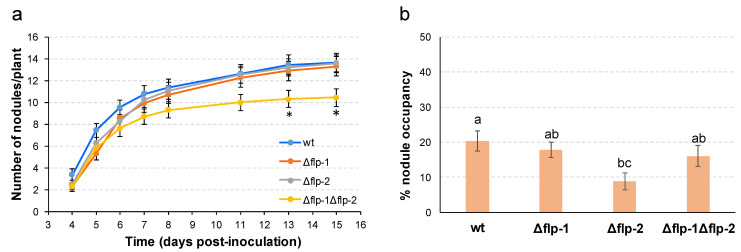
Symbiotic phenotype of GR4-derivative *flp* deletion mutants. (**a**) Nodulation kinetics of alfalfa plants inoculated with *S. meliloti* wild-type GR4 or their derivative single and double *flp* deletion mutants. Data represent the average number of nodules per plant. The bars represent standard errors from 38 replicates in two independent experiments. Asterisks indicate significant differences compared to the wild-type according to an ANOVA test (*p* ≤ 0.05). (**b**) Competitive nodulation assays. Data represent the percentage of nodules occupied by each rhizobial strain after co-inoculation with the marked strain GR4 (pGUS3) in a mixture ratio of 1:1. Error bars indicate the standard error of the mean obtained from three independent experiments. Different letters indicate significant differences according to an ANOVA test (*p* ≤ 0.05).

## Data Availability

Data are contained within the article and supplementary materials.

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
