# Peer review of "Sinorhizobium meliloti GR4 Produces Chromosomal- and pSymA-Encoded Type IVc Pili That Influence the Interaction with Alfalfa Plants"

_plants, 2024, doi:10.3390/plants13050628_

Round 1

Reviewer 1 Report

Comments and Suggestions for Authors

Article 1

Sinorhizobium meliloti GR4 produces chromosomal- and 2 pSymA-encoded Type IVc Pili that influence the interaction 3 with alfalfa plants PRESENTS USEFUL INFORMATION on symbiotic bacteria of alfalfa, an economic  plant with enormous application potential. the manuscript is well written and presented. Please, check for hypothesis, controls and references.

Comments on the Quality of English Language

Article 1

Sinorhizobium meliloti GR4 produces chromosomal- and 2 pSymA-encoded Type IVc Pili that influence the interaction 3 with alfalfa plants PRESENTS USEFUL INFORMATION on symbiotic bacteria of alfalfa, an economic  plant with enormous application potential. the manuscript is well written and presented. Please, check for hypothesis, controls and references.

Author Response

We would like to thank the reviewer for helpful and constructive comments which without a doubt have helped us to improve the manuscript.

Response to Reviewer Comments:

Point 1: Sinorhizobium meliloti GR4 produces chromosomal- and 2 pSymA-encoded Type IVc Pili that influence the interaction 3 with alfalfa plants PRESENTS USEFUL INFORMATION on symbiotic bacteria of alfalfa, an economic plant with enormous application potential. the manuscript is well written and presented. Please, check for hypothesis, controls and references.

Response 1: Thank you for the positive comments. Hypothesis, controls and references have been checked.

Reviewer 2 Report

Comments and Suggestions for Authors

This manuscript describes the genetic function of the pili gene cluster of Sinorhizobium meliloti, but I cannot understand its importance of plant nodulation symbiosis. In addition, I believe there are some important concerns that the author should address.

What's the difference from S. meliloti Rm1021 and GR4? Since Rm1021 is a model strain, why not the authors constructed the deletion mutant in the background?

The information from Figure 1 and Table 1 can be merged together.

Figure 2 is redundant and of little significance.

Figure S5 should be placed in the main text in the form of a graph, clearly informing readers of the missing mutation information.

The scale of Figure 3 is chaotic, why can't it be unified into two scales.

How to obtain the data for Figures 4b and d? The results of c and d are inconsistent.

The adhesion and cloning ability of flp-1 and flp-2 double deletions were enhanced (Fig.5), but the tumor formation ability was weakened (Fig.6a), while the tumor occupancy rate remained unchanged (Fig.6b), which is contradictory. How can we explain this?

Does the double deletion mutant of flp-1 and flp-2 affect the growth of alfalfa after inoculation?

Author Response

We would like to thank the reviewer for helpful and constructive comments which without a doubt have helped us to improve the manuscript. 

Response to Reviewer Comments:

Point 1: This manuscript describes the genetic function of the pili gene cluster of Sinorhizobium meliloti, but I cannot understand its importance of plant nodulation symbiosis.

Response 1: S. meliloti T4cP do not play an essential role in the establishment of symbiosis since mutants lacking one or both flp clusters are still capable of inducing nitrogen-fixing nodules in alfalfa plants. However, our data demonstrate that T4cP influence the symbiotic fitness of S. meliloti by affecting i) bacterial attachment to plant roots (Figure 5), ii) the total number of root nodules developed by the plant (Figure 6a), and iii) the bacterial competitive ability for nodule occupancy (competitiveness) (Figure 6b). These traits, although not essential for symbiosis, can impact rhizobial fitness in natural soil environments where bacterial cells must compete with other plant-interacting microorganisms including other rhizobia.

For more clarity on the importance of the symbiotic phenotypes associated to our flp mutants we have included additional information and references in the Introduction section (pages 1-2, lines 41-46).

Point 2: In addition, I believe there are some important concerns that the author should address. What's the difference from S. meliloti Rm1021 and GR4? Since Rm1021 is a model strain, why not the authors constructed the deletion mutant in the background?

Response 2: Although less investigated than the well-known strain Rm1021, GR4 is also a model strain used to study different aspects of the S. meliloti biology. Investigations on GR4 (more than 30 publications according to PubMed) have shown that this strain exhibits several relevant differences compared with Rm1021. A recent publication in which different S. meliloti strains were used, including GR4 and Rm1021, revealed GR4 as one of the strains showing the highest values of nitrogenase activity and plant growth promotion as well as a highly competitive strain for nodulation of alfalfa (Bellabarba et al. 2021 [13]). In competition with Rm1021, GR4 was the best competitor occupying 93.4% of the nodules (Fig. 1A in Bellabarba et al. 2021 [13]). The molecular bases responsible for the high competitive ability of GR4 are complex and not fully understood (Bellabarba et al. 2021 [13]). In our previous investigations, we demonstrated that GR4 is more efficient than Rm1021 in developing biofilms on abiotic and plant root surfaces, which could account for the better symbiotic performance of GR4 (Amaya-Gómez et al. 2015 [9]). Therefore, we decided to investigate the role of the flp clusters and T4cP in the root colonization ability and symbiotic behavior of GR4.

In the new version of our manuscript, we have included additional information about GR4 and we have extended the explanations on the rationale of studying the flp clusters in this S. meliloti strain (page 3, lines 120-128).

Point 3: The information from Figure 1 and Table 1 can be merged together.

Response 3: We find it impossible to merge Figure 1 and Table 1 without losing relevant information. Figure 1 shows the genomic context of tad/cpa/ctp loci in 6 different bacterial species and easily allows to graphically identify similarities in gene organization as well as putative orthologues and paralogues. Table 1 provides different and additional information on: i) Gene identifiers for each locus in GR4 and Rm1021 (in Figure 1, only names and gene identifiers for Rm1021 were used for short word length); ii) the putative function of the products of different T4cP loci in S. meliloti; and iii) the degree of similarity as percent of amino acid sequence identities between each S. meliloti gene and their closest orthologues in other bacteria. Therefore, we believe that Figure 1 and Table 1 should be kept separated.

Point 4: Figure 2 is redundant and of little significance.

Response 4: The aim of this Figure was to show: i) the truncated structure of the Tad secretion system encoded by the flp-2 cluster, which contrasts with the complete T4cP apparatus encoded by the chromosomal flp-1 cluster; and ii) the apparent complementarity of the proteins encoded by flp-2 and Locus 3 to build a complete Tad secretion system. Considering the reviewer’s comment, we have removed Figure 2 from the main manuscript. However, to help those readers who are not very familiar with T4cP and need more information about the protein composition and structure of Tad secretion systems, we have included the figure in the Supplementary Materials (Figure S1).

Point 5: Figure S5 should be placed in the main text in the form of a graph, clearly informing readers of the missing mutation information.

Response 5: Old Figure S5 has been moved to the main manuscript (now Figure 2). Information on the genes that were removed in the deleted version of the flp-1 and flp-2 clusters has been included in the corresponding figure legend.

Point 6: The scale of Figure 3 is chaotic, why can't it be unified into two scales.

Response 6: We apologize for not having better unified the scale in this figure. To improve it, panels i and j have been replaced in the new Figure 3 so that now, only two scales for the higher magnification and two scales for the lower magnification are shown.

Point 7: How to obtain the data for Figures 4b and d?

Response 7: Data shown in Figure 4b representing the average of surface migration was determined as the average length of the two sides of a rectangle able to exactly frame each colony obtained in MM (0.6% agar) after 48 h of incubation at 28 °C. Means and standard errors were obtained from at least 2 replicates in three independent experiments. To obtain data shown in Figure 4d the diameter of the swimming halo developed by each colony in BM (0.3 % agar) was measured after 96 hours of incubation at 28ºC. Means and standard errors were obtained from 3 replicates. In the new version of our manuscript, the information about the experimental details have been improved in Materials and Methods section (pages 16-17, lines 621-633), including the correction of the incubation time used for determining the swimming halo (96h and not 72h). The number of samples used for the measurements are now indicated in the figure legend. Moreover, during the revision of this figure, we detected a mistake in the letters used to indicate statistical significance of data shown in the graph of panel b. The mistake, that does not affect the conclusions described in the manuscript, has now been corrected in the new Fig. 4. We apologize for the mistakes and thank the reviewer for the comments on Figure 4.

Point 8: The results of c and d are inconsistent.

Response 8: The results shown in panels c and d of Fig. 4 might look inconsistent but they are not. The differences in the swimming halo diameter are just 1 to 2 mm, a difference which is difficult to perceive just by looking at the representative picture. For clarity and also to address Point 1 of Reviewer #3, in the new Fig. 4 we have increased the brightness and contrast equally in all photos, so that the colonies can be seen more clearly.

Point 9: The adhesion and cloning ability of flp-1 and flp-2 double deletions were enhanced (Fig.5), but the tumor formation ability was weakened (Fig.6a), while the tumor occupancy rate remained unchanged (Fig.6b), which is contradictory. How can we explain this?

Response 9: Firstly, we would like to clarify that although the double flp-1flp-2 mutant shows greater adhesion to plant roots than the wild-type strain (indicated by the number of CFUs recovered from roots 2 hours after inoculation), the root colonization (“cloning”) ability was not enhanced in the mutant. As shown in Figure 5 and explained in the manuscript (page 11 lines 357-359 in the Results section and page 15, lines 528-532 in the Discussion section), the double flp-1flp-2 mutant colonized alfalfa roots as efficiently as the wild-type as indicated by the recovery of similar numbers of CFUs 24, 48 and 72 hours post-inoculation.

The double flp-1flp-2 mutant, which exhibits increased adhesion to plant roots but is unaffected in root colonization and competitiveness, induces a lower number of nodules on alfalfa roots than the wild-type strain. As indicated in the Discussion section (page 15, lines 536-537), the reasons for the lower nodule formation performance of the double mutant are unknown and we can only speculate. Two non-excluding possibilities have been suggested in the Discussion section (page 15, lines 537-546): 1) Increased attachment to the root surface hampers the bacterial ability to move and timely find proper infection sites, and/or 2) bacterial cells that lack T4cP induce plant defense responses that restrict the progression of further infection events. The observation that the double mutant induces a lower number of nodules in single inoculation experiments but exhibits similar competitive ability for nodule occupancy when co-inoculated with the wild-type are not necessarily contradictory results. As commented above, competitiveness is a complex trait. The outcome of nodule occupancy determined in competition assays cannot be predicted from data obtained in single inoculation experiments (Granada Agudelo et al. 2023 [14]). As an example, a theoretical rhizobial strain capable of producing a bacteriocin might induce a low number of nodules in single inoculation experiments but occupy 100% of the nodules when co-inoculated with a bacteriocin-sensitive rhizobial strain.

To help the reader in the understanding of these complex phenotypes, additional information and references have been included in the new version of our manuscript in the Introduction section (pages 1-2, lines 41-46) and in the Discussion (page 15 lines 540-544).

Point 10: Does the double deletion mutant of flp-1 and flp-2 affect the growth of alfalfa after inoculation?

Response 10: No. The growth of alfalfa plants inoculated with the double deletion mutant was similar to that of plants inoculated with the wild-type strain. We have included a representative image in the Supplementary Material (Figure S8).

Reviewer 3 Report

Comments and Suggestions for Authors

The manuscript reporting experiments and findings on the phenotype of deletion mutants of two of the pilus-encoding regions, or both is impeccably written. I have items requiring attention:

1: While the figures in the supplemental are very high quality, the photos in Fig 4 of the paper itself are of insufficient clarity. Please upload images where colonies can be clearly seen.

2: The competitive modulation assay as described makes no sense to me. If I read the Methods and figure 6b legend correctly, the first bar on the left of Fig 4b represents wild type. These plants appear to have been exposed to wt GR4 and GR4 with pGUS3. Does a result of 20% imply that the pGUS3-bearing strain was four times as competitive as the wt without plasmid? If I have misunderstood the methods description, then other readers may also, so please expand the methods to help prevent misunderstanding. If my interpretation is correct, then the competition assay is stacked against the wild type and therefore possible its derivative mutants.

Line 259: Change "difficulting" to "making complementation experiments more difficult."

Author Response

We would like to thank the reviewer for helpful and constructive comments which without a doubt have helped us to improve the manuscript.

Response to Reviewer Comments:

Point 1: While the figures in the supplemental are very high quality, the photos in Fig 4 of the paper itself are of insufficient clarity. Please upload images where colonies can be clearly seen.

Response 1: We apologize for the insufficient clarity of the photos in the previous Fig. 4. In the new Fig. 4 we have increased the brightness and contrast equally in all photos so that the colonies can be seen more clearly.

Point 2: The competitive modulation assay as described makes no sense to me. If I read the Methods and figure 6b legend correctly, the first bar on the left of Fig 4b represents wild type. These plants appear to have been exposed to wt GR4 and GR4 with pGUS3. Does a result of 20% imply that the pGUS3-bearing strain was four times as competitive as the wt without plasmid? If I have misunderstood the methods description, then other readers may also, so please expand the methods to help prevent misunderstanding. If my interpretation is correct, then the competition assay is stacked against the wild type and therefore possible its derivative mutants.

Response 2: The interpretation of the Methods by the reviewer is correct but not the interpretation of the results. Indeed, the first bar on the left of Fig. 4b represents the percent of white nodules (20%) occupied exclusively by the unmarked GR4 strain in the control inoculation mixture GR4 vs GR4pGUS3. The remaining nodules (80%) were blue and correspond to nodules occupied by the marked strain GR4 (pGUS3) but also nodules with double occupancy (occupied by the unmarked and marked strains), which also stain blue. The percent of double occupancy cannot be determined with the methodology used in our competition assays. Therefore, we compare the percentage of white nodules obtained in the different inoculation mixtures. Significant differences with respect to the percentage of white nodules obtained in the control inoculation mixture indicate differences in competitiveness between GR4 and the tested strain. To avoid misunderstandings, we have extended the information on the competition assays in the Materials and Methods section (page 17, lines 656-664).

Point 3: Line 259: Change "difficulting" to "making complementation experiments more difficult."

Response 3: The change has been made (page 8, lines 266-267).